# SIVA: Self Improving Vulnerability Agent

**Valentin Walischewski**
Imperial College London
London, UK
`vtw23@ic.ac.uk`

**Giulio Zizzo**
IBM Research Europe
Dublin, Ireland
`giulio.zizzo2@ibm.com`

**Kevin N. Webster**
Imperial College London
London, UK
`knw@ic.ac.uk`

## Abstract

In the ever more digitalized world of today, code vulnerabilities pose a critical threat to our privacy, economy, safety, and infrastructure. Existing automated code vulnerability detection methods suffer from high false positive rates, poor generalization and their inability to adapt to changing vulnerability landscapes. To address these challenges we propose SIVA, a self-improving LLM-based vulnerability detection agent, using memory-guided meta-learning for dynamic prompt optimization. SIVA showed strong learning capabilities, improving its F1 score from 58% to 95% in 5 iterations, significantly outperforming previous state-of-the-art multi-agent systems ($\approx 53\%$ F1) on real-life vulnerability datasets. Furthermore, SIVA generalized well across 7 programming languages (93% F1), successfully transferring learned vulnerability concepts between them.[1]

## 1 Introduction

Code Vulnerabilities present an ever-growing and extremely critical threat in the rapidly digitalizing world of today. Vulnerabilities often involve subtle deviations from the safe code, but their impact is severe and real. This is highlighted by the recent identification and subsequent exploitation of a set of vulnerabilities in Microsoft's SharePoint system, putting many large enterprises and institutions at high risk (HackerOne, 2024; Unit 42, 2025; ZeroThreat, 2025).

Facing this crisis and aware of the scarcity of human security experts, the last decade has seen a surge in research interest for building automated code vulnerability detection systems. These systems rely on detailed static analysis (SA) or machine learning (ML) techniques, such as Deep Learning (DL), and whilst successful for some vulnerability categories or domains both approaches face inherent limitations. Charoenwet et al. (2024) conducted a comprehensive study of current state-of-the-art static analysis tools, reporting a false positive rate (FPR) of 76% and a false negative rate (FNR) of 22%. Although DL-based methods show promising potential, Chakraborty et al. (2024) showed that these models often suffer from severe overfitting, showing poor performance on realistic test data.

In addition, recent work has investigated the feasibility of using LLMs in code vulnerability detection. Ullah et al. (2024) found that when evaluating LLM's on real-world vulnerability data, they only achieved a reasoning adjusted F1 score of 14%, showcasing the poor understanding of the models. Furthermore, investigations into the use of agents for vulnerability detection revealed that even sophisticated LLM systems struggle with this complex task (Ahmed et al., 2025; Yildiz et al., 2025).

All of these traditional approaches are static in nature and cannot continuously learn to adapt to new vulnerabilities or subtle changes. Secondly, current approaches only allow limited knowledge acquisition and utilization, relying predominantly on information included in their training data or pattern library. To address these challenges, we propose a novel vulnerability detection framework, **SIVA** (Self-Improving-Vulnerability-Agent). Inspired by recent advances in self-improving LLM-based agentic systems (Hu et al., 2025; Zhang et al., 2025b; Robeyns et al., 2025), SIVA is, to the best

---

[1]The code is available at: https://github.com/Valliwa/SIVA

39th Conference on Neural Information Processing Systems (NeurIPS 2025) Workshop: .

of our knowledge, the first self-improving LLM agent for the cybersecurity domain, achieving state-of-the-art results across 7 different programming languages, when evaluated on real-life vulnerability datasets. SIVA's architecture is grounded in information-theoretical intelligence evolution insights and inspired by the analogy of human problem solving, where experience and solution transfers play an essential role. Similar to humans, SIVA improves through practice and exposure to diverse examples, by building and utilizing a comprehensive knowledge-base, containing learned patterns and successful strategies (Liu et al., 2025a; Liu and van der Schaar, 2025; Hu et al., 2025).

Our contributions in this paper are:

- SIVA, the first self-improving LLM-based vulnerability detection agent that achieves state-of-the-art results (95% F1 score) through memory-guided meta-learning and dynamic prompt optimization, significantly outperforming existing multi-agent systems (53% F1 score) on real-world vulnerability datasets.
- Novel learning dynamics in self-improving systems, showing that SIVA successfully transfers learned vulnerability concepts across 7 programming languages (93% F1 score) and develops conceptual understanding through hierarchical reasoning strategies inspired by information-theoretic intelligence evolution.

This paper proceeds as follows: Section 2 will discuss related work in the fields of automated vulnerability detection and self-improvement agentic systems. Secondly, in Section 3, we will present and discuss the proposed SIVA framework and architecture. Then in Section 4 we present detailed evaluation results and discuss their implications for SIVA's learning capabilities. Finally, in Section 5, we will conclude on our findings and highlight potential areas for future research.

## 2 Related Work

**Code Vulnerability Detection:** Static analysis tools (SA) for vulnerability detection rely essentially on hardcoded patterns for well-known vulnerabilities, leading to poor generalization, single language dependencies, and technical challenges during practical adoption (Wadhams et al., 2024; Charoenwet et al., 2024).

More modern deep learning systems based on graph learning like VulDeePecker show promising performance, significantly reducing FNR with more acceptable FPR levels (Yamaguchi et al., 2014; Li et al., 2018; Liu et al., 2025b). However, a recent evaluation of these methods on more realistic datasets showed that they severely suffer from overfitting and poor generalization (F1 score and precision decrease by 91% and 95%, respectively, on realistic data) (Chakraborty et al., 2024; Lu et al., 2024).

Following the consensus that SOTA LLMs alone are not yet capable of reliable code vulnerability detection (Ullah et al., 2024; Yildiz et al., 2025), more recent work has looked at various ways to enhance LLM-based detection systems. Li et al. (2025) found that context inclusion can significantly improve performance, whilst agent or multi-agent frameworks showed only little performance improvements, failing to compensate for the underlying LLM's incapability (Ahmed et al., 2025; Yildiz et al., 2025). In addition to that, the integration of graph learning with LLMs has emerged as a promising direction, showing improved generalization and outperforming existing DL methods (Chu et al., 2024; Lu et al., 2024; Liu et al., 2025b).

**Self-Improving Systems:** Recent advances in LLM performance and reasoning have led to the development of LLM-based foundation agents (Naveed et al., 2024; Liu et al., 2025a; Chen et al., 2025). An LLM-based foundation agent is defined as an autonomous, adaptive intelligent system that can actively perceive various environment signals, learn from experience to update its structured internal states (memory, world model, goal, etc.), reason (about its internal and external states), and take actions to accomplish complex long-term objectives.

The manual design of agents or multi-agent systems is a highly complex and labor intensive task that requires significant effort and expertise. Therefore, more recent research has focused on the development self-improving agentic systems. Beyond optimizing the underlying model weights, new approaches focus primarily on: i) prompt optimization, ii) workflow optimization, iii) tool use optimization, and iv) architecture optimization (Zhang et al., 2025b; Liu et al., 2025a).

Hu et al. (2025) proposed the Automated Design of Agentic Systems (ADAS) framework, where a separate meta-agent can alter an underlying agentic systems architecture based on environment feedback, inventing and testing new components and component interaction mechanisms. Extending this work, (Robeyns et al., 2025) and (Yin et al., 2025) developed agent systems that directly include meta-learning and code-based improvement capabilities, simulating "self-awareness" and achieving remarkable learning in various tasks, as well as SOTA performance across various evaluations.

Liu et al. (2025a) defined an information theoretical measure of the agent's intelligence $IQ_t^{agent}$, at time $t$:

$$IQ_t^{agent} \equiv -D_K(\theta, M_t^{memory}) = \sum_{\mathbf{x} \in \mathcal{U}} P_{\mathcal{W}}(\mathbf{x} \mid M_t^{memory}) \log\{\frac{P_{\mathcal{W}}(\mathbf{x} \mid M_t^{memory})}{P_{\theta}(\mathbf{x} \mid M_t^{memory})}\} \quad (1)$$

where $\mathcal{U}$ represents the unknown knowledge of the world, defined as $\mathcal{U} = \mathcal{W} \setminus M_t^{\text{memory}}$, i.e. all knowledge of the world $\mathcal{W}$ excluding what is stored in SIVA's memory, $M_t^{\text{memory}}$. Therefore, $IQ_t^{agent}$ measures both the amount of existing knowledge and the efficiency of its application. Taking the expected value with respect to newly acquired knowledge, $\mathbf{x}_\Delta \sim \mathcal{U}$, Liu et al. (2025a) then derived the following expression for expected intelligence at time $t = t + 1$:

$$\mathbb{E}_{\mathbf{x}_\Delta \sim \mathcal{U}}[IQ_{t+1}^{agent}] = IQ_t^{agent} + \sum_{\mathbf{x} \in \mathbf{x}_\Delta} P_{\mathcal{W}}(\mathbf{x} \mid M_t^{memory}) \log\{\frac{P_{\mathcal{W}}(\mathbf{x} \mid M_t^{memory})}{P_{\theta}(\mathbf{x} \mid M_t^{memory})}\} \quad (2)$$

Here, the second term represents the relative entropy of the conditional probability distribution of $\mathbf{x}_\Delta$, conditioned on $M_t^{memory}$, which is always non-negative. Henceforth, from equation 2 we can conclude that for an agent, equipped with memory, $M_t^{memory}$, and parametrized by $\theta$, its intelligence is a non-decreasing function, w.r.t. its acquired knowledge. In addition to that, equation 2 also shows that the expected intelligence gain at time $t + 1$ depends on the KL discrepancy between the models $(P_{\theta}(\mathbf{x} \mid M_t^{memory}))$ and the actual distribution $(P_{\mathcal{W}}(\mathbf{x} \mid M_t^{memory}))$. This implies that the rate of intelligence growth is highest when new knowledge is most diverse or unexpected.

## 3 Self Improving Vulnerability Agent

We now introduce and detail SIVA. We start by providing the more precise definition,

> ***Definition 3.1 - SIVA***: A foundation agent equipped with an improved mental state that allows it to iteratively self-improve its vulnerability detection capabilities, through meta-learning (Hu et al., 2025), dynamic prompt evolution (Zhang et al., 2025b), and CWE-specific pattern recognition mechanisms (Liu et al., 2025b).

Formally, we express SIVA as a foundation agent with enhanced mental state (Liu et al., 2025a):

$$a_t^{(\text{SIVA})} = \pi_{\text{SIVA}}(o_t, s_t, M_t^{(\text{SIVA})}, \theta_t) \quad (3)$$

Where: $a_t^{(\text{SIVA})}$ is the vulnerability analysis - expressed as action, $\pi_{\text{SIVA}}$ is the meta-learning agent policy, $o_t$ is an observation, $\theta_t$ are the LLM parameters, and $M_t^{(\text{SIVA})} = \{M_t^{(\text{memory})}, M_t^{(\text{meta})}, M_t^{(\text{prompts})}, M_t^{(\text{patterns})}\}$ is the enhanced mental state, including:

- $M_t^{(\text{memory})}$, the agents memory system
- $M_t^{(\text{meta})}$, the strategy weights (obtained through meta-learning)
- $M_t^{(\text{prompts})}$, the dynamic prompt library
- $M_t^{(\text{patterns})}$, the CWE-specific pattern analysis state

This allows SIVA to recall its previous analysis attempts and learn from them to improve its performance over time, essentially creating a large database of worked solutions and failed attempts to draw insights and generalizations. Analogously to human learning, SIVA builds its understanding by recognizing solution patterns across vulnerability types, and adapting its knowledge to novel contexts (Liu and van der Schaar, 2025; Liu et al., 2025a).

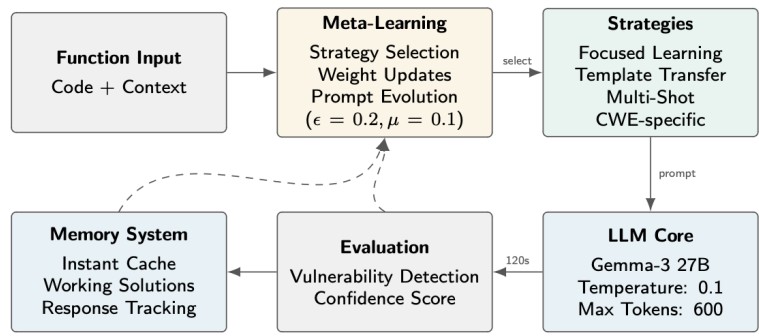

Figure 1: SIVA Architecture: A schematic visualization of the full self-improvement process and its components. Solid lines represent the data flow, dashed lines represent internal feedback loops.

## 3.1 Memory Architecture

Inspired by human learning patterns, SIVA's memory architecture balances computational efficiency with learning effectiveness. Its memory consists of a three-tier system with: i) instant cache, allowing immediate recall for exact matches, similar to humans, who, when remembering an answer, can recognize solutions instantly. ii) Retrieval of working solutions based on patterns for similar problems, reflecting the human approach of constructing novel solutions from their knowledge base of what works and what does not (Hu et al., 2025; Yin et al., 2025). iii) Temporal response tracking for strategy optimization, mirroring humans, who can constantly update their solution approaches, based on new insights gained during analysis or practice (Zhang et al., 2025b).

**Instant Cache:** Given that in both an experimental setup and real-life vulnerability detection settings we often face duplicates of the exact same function, we incorporated an instant look-up system, which can return the successful analysis for a previously analyzed function in $\mathcal{O}(1)$ time, by deploying SHA-256 hashing (Kashmar, 2024), (Appendix A.14.4).

Each function is given a unique lookup value, $\text{hash}(f_i)$, and if a successful high-quality solution, $a_i^*$ exists in SIVA's set of all successful solutions, $\mathcal{S}_{suc}$, the cache mechanism immediately returns the exact solution, $a_i^*$ for $\text{hash}(f_i)$. In order to prevent guessed or poor quality solutions from poisoning the learning process, we ensure that all solutions exceed a quality threshold hyperparameter, $\forall a_i^* \in \mathcal{S}_{suc}, a_i^* > \tau_{\text{quality}}$. The quality of an answer is quantified using the following measure (Appendix A.14.1):

$$Q(a_i) = \alpha \cdot \text{keyword\_density} + \beta \cdot \text{analysis\_depth} + \gamma \cdot \text{structural\_correctness} \qquad (4)$$

Where, we used sets of predefined keywords to assess these quality factors.

**Working Solutions Repository:** Given that vulnerability patterns, such as for example cross-site scripting, repeat across different, yet structurally similar functions, SIVA can adapt existing successful approaches for new, but similar functions, contrary to existing methods treating detection tasks in isolation (Ullah et al., 2024; Chakraborty et al., 2024).

Based on the principle of analogical reasoning for humans (Liu et al., 2025a), SIVA maintains a structured library of successful solution approaches, enabling it to leverage and adapt previously solved cases (e.g. buffer-overflows) to novel contexts. The library, $\mathcal{W} = \{(\rho_i, s_i^*) : \rho_i \in \mathcal{P}, s_i^* \in \mathcal{S}_{suc}\}$, is indexed by vulnerability patterns $\rho_i \in \mathcal{P}$ (e.g. unvalidated pointer arithmetic, privilege escalation vectors) in the set of all stored patterns $\mathcal{P}$. Therefore, SIVA can generalize vulnerability concepts, recognizing that vulnerability patterns share conceptual similarities between languages (e.g. python deserialization and C/C++ buffer overflows), and successfully using these working solutions, $s_i^*$ for novel contexts.

SIVA stores the full analysis and reasoning for each solution, $\mathcal{W}[\rho_i]$, not just the binary detection results. This enables the agent to successfully utilize its accumulated memory, to improve its intuition

and evolve from basic funtion-level matching to sophisticated pattern matching and conceptual abstraction. For example, a Cross-site Scripting (CWE-79) template might initially focus on simple pattern matching for script tags, but evolve to understand complex encoding schemes, DOM manipulation, and framework-specific sanitization bypasses.

**Response Evolution Tracking:** SIVA maintains temporal records of its analysis attempts, capturing how its CWE specific detection strategies evolve over time. This mechanism is inspired by metacognitive learning, enabling SIVA to reflect on its own learning process, identifying improvement strategies or areas of stagnation (Liu and van der Schaar, 2025). The underlying process resembles second-order differential calculus applied to text space optimization, allowing the agent to escape local performance maxima (Zhang et al., 2025b). The temporal data also shows CWE specific learning trajectories, informing SIVA's decision, whether to persist with the curent learning strategy or whether to explore alternatives.

For each function, $f_i$, SIVA tracks the progression of its reasoning quality (equation 4), error patterns, and refinements, by storing function-specific response sequences, $\mathcal{R}_i = \{e_t^{(i)} : t \in [t, T_i]\}$ in its memory.

## 3.2 Learning Strategies

Inspired by mathematical problem solving, SIVA can use three complementary learning strategies, each suitable for a certain level of novelty and complexity. Together, they form a hierarchical reasoning system that can evolve from basic prior-knowledge application (i.e. LLM weights) to sophisticated analogical reasoning and diversity enhanced multishot learning (Hu et al., 2025; Liu et al., 2025a; Liu and van der Schaar, 2025).

**Focused Learning (FL):** If SIVA has successfully analyzed a similar function before, it will adapt that analysis to the new function. This most direct from of knowledge transfer, mirror how a human analyst would immediately recognize familiar vulnerability classes and apply proved detection approaches. Instead of performing its analysis from scratch, SIVA leverages its memory of successful reasoning patterns, to customize proven analytical structures, in order to apply them to novel contexts.

Given a function $f_i$ and at least one existing working solution, $\mathcal{W}[\rho_i]$, for the respective vulnerability pattern, $\rho_i = \rho(f_i)$, SIVA will then adapt a focused learning prompt template, $\mathcal{T}_{\text{focused}}$, using the working solutions, $\mathcal{W}[\rho_i]$, and function data, $f_i$, to create a new prompt for the LLM to process.

$$P_{\text{focused}}(f_i, \rho_i) = \mathcal{T}_{\text{focused}}(\mathcal{W}[\rho_i], f_i) \tag{5}$$

The resulting prompt, $P_{\text{focused}}(f_i, \rho_i)$, is then passed to the LLM for analysis.

**Template Transfer (TT):** If there are no pattern-specific solutions in SIVA's memory, $s_i^* \notin \mathcal{S}_{suc}$, it then looks for successfully analyzed functions from the same CWE family, $\gamma$, to transfer potentially valuable insights or patterns to the analysis of $f_i$. TT is based on the observation that although syntactically different, vulnerabilities in the same conceptual family often share underlying principles (e.g. buffer-overflow, CWE-119, and out-of-bounds writes, CWE-787, both require boundary analysis.)

$$P_{\text{template}}(f_i, \gamma) = \mathcal{T}_{\text{template}}(\text{argmax}_{s \in \mathcal{S}_\gamma} Q(s), f_i) \tag{6}$$

Where, $Q(s)$ represents the analysis quality (Equation 4). The constructed prompt, $P_{\text{template}}(f_i, \gamma)$, is then passed to the LLM, including both the successful example and the new function code.

**Multi-shot Learning (MSL):** For challenging functions, where there exist multiple successful attempts within the same CWE family, SIVA can gather multiple examples to learn from. SIVA constructs robust detection strategies, by balancing example quality, $Q(s)$, with analytical diversity, $D(s)$. This ensures sufficient conceptual coverage, whilst preventing overfitting (Liu et al., 2025a).

$$P_{multishot}(f_i, \gamma) = \mathcal{T}_{multishot}(\text{DiverseRecent}(S_\gamma, k), f_i) \tag{7}$$

Where DiverseRecent($S_\gamma, k$) returns up to $k$ most recent successful examples from $\mathcal{S}_\gamma$ with distinct vulnerability patterns (Appendix A.14.2).

## 3.3 Meta-Learning

Different problems, in different contexts, require different learning strategies (Li et al., 2025; Zhang et al., 2025a; Liu et al., 2025a). SIVA applies meta-learning to its accumulated performance data $\mathcal{R}$, to learn which approach is best for each type of vulnerability. We express this method as a contextual bandit approach for optimal strategy selection (Sutton and Barto, 2018; Liu et al., 2025a).

This process was inspired by human analysts and their ability to develop their intuition of which strategy is optimal for a specific vulnerability class. It represents SIVA's metacognitive ability to learn how to learn (Liu and van der Schaar, 2025).

**1. Strategy Weight Evolution** SIVA deploys a dynamic strategy weighting system, which continuously adapts to new empirical insights. Similar to multiarmed bandit settings, SIVA must balance the exploitation of proven approaches with the exploration of potentially superior strategies. In order to optimally balance recent performance data with historical insights, the agent leverages an exponential moving average model (Sutton and Barto, 2018):

$$w_{t+1}^{(\xi)} = (1 - \mu) \cdot w_t^{(\xi)} + \mu \cdot \text{SuccessRate}_t(\xi) \tag{8}$$

Where: $\xi \in \Xi = \{\text{cache}, \text{FL}, \text{TT}, \text{MSL}, \text{CWE\_specific}\}$ is a strategy in SIVA's set of available strategies $\Xi$, $w_t^{(\xi)}$ is the weight of strategy $\xi$ in iteration $t$, $\mu$ is the learning rate (adaptability vs. stability), and $\text{SuccessRate}_t(\xi) = \frac{\text{Successes}(\xi)}{\text{Attempts}(\xi)}$ is the success rate of the strategy $\xi$ up to iteration $t$.

This adaptive weighting ensures that SIVA gradually favors strategies that demonstrate consistent success while maintaining sufficient flexibility to adapt to new vulnerability patterns. For example, if FL consistently achieves $95\%$ success on buffer overflows while TT only achieves $60\%$, the weights will evolve to strongly prefer FL for similar functions.

**2. Meta-Decision Function** SIVA uses a $\epsilon$-greedy strategy that balances exploration of new strategies with exploitation of proven approaches. This mechanism prevents SIVA from becoming trapped in local maxima, while ensuring efficient use of successful strategies (Sutton and Barto, 2018).

$$\xi^* = \begin{cases} \text{explore}(\Xi) & \text{if } \chi < \epsilon \\ \text{argmax}_{\xi \in \Xi} \{w_t^{(\xi)}\} & \text{otherwise} \end{cases} \tag{9}$$

Where, $\chi \sim \text{Uniform}(0, 1)$, $\epsilon$ is the exploration rate hyperparameter.

**3. Prompt Evolution** Existing LLM-based vulnerability detection systems rely on static prompts, which often fail to adapt to highly specific cases or novel vulnerability patterns (Ullah et al., 2024; Li et al., 2025). SIVA is able to create and maintain a library of CWE-specific prompts that evolve over time based on their performance. Therefore, SIVA can develop increasingly sophisticated domain expertise for specific CWE families.

Given a function, $f_i$, that belongs to a CWE family $\gamma(f_i)$, SIVA can store and evolve a specific template tuple $\gamma, T^{(\gamma)}$, (Zhang et al., 2025b).

$$T^{(\gamma)} = (\gamma, \tau^{(\gamma)}, \sigma^{(\gamma)}, \nu^{(\gamma)}, \lambda^{(\gamma)}, \kappa^{(\gamma)}, \phi^{(\gamma)}) \tag{10}$$

Where: $\gamma$ is the CWE identifier (e.g. CWE-384), $\tau^{(\gamma)}$ is the prompt template (Appendix A.3.1), $\sigma^{(\gamma)}$ is the template success rate, $\nu^{(\gamma)}$ is the usage count, $\lambda^{(\gamma)}$ is the "last updated" timestamp (for temporal tracking), $\kappa^{(\gamma)}$ represents a set of insights from relevant successful analyses, and $\phi^{(\gamma)}$ represent the template specific failure patterns.

The prompt evolution process follows a systematic refinement cycle:

1. **Performance Monitoring:** After each analysis attempt, SIVA evaluates whether the template correctly identified the vulnerability, tracking both the success and failure modes.

2. **Insight Extraction:** Succesful analyses contribute new detection patterns, enriching the underlying knowledge base (e.g discovering that unescaped user input in dynamic SQL queries indicates SQL injection vulnerability).

3. **Failure Analysis:** Failed detections allow the identification of systematic blind spots in $\phi^{(\gamma)}$, for example, if multiple SQL injection vulnerabilities are missed due to prepared statement misuse, this pattern is incorporated into the template.

4. **Template Refinement:** Based on accumulated insights and failure patterns, SIVA refines $T^{(\gamma)}$, to include novel detection logic, remove ineffective approaches, and optimize the analytical strategy. This refinement process uses natural language generation (Google DeepMind, 2024) to reformulate prompts while preserving successful elements.

5. **Evolutionary Selection:** Templates with consistently low success rates, $\sigma_\gamma < \Phi_{\text{evolve}}$, undergo major restructuring or replacement, while high-performing templates are preserved and only gradually refined (Appendix A.14.5).

## 3.4 Learning Dynamics

SIVA's memory-guided meta-learning framework ensured that its intelligence acquisition followed the equation 2, which we applied to SIVA, obtaining the following expression:

$$
\mathbb{E}_{\mathbf{f}_\Delta \sim \mathcal{U}}[IQ_{t+1}^{(SIVA)}] = IQ_t^{(SIVA)} + \sum_{f_i \in \mathbf{f}_\Delta} P_{\mathcal{W}}(f_i \mid M_t^{(SIVA)}) \log\{\frac{P_{\mathcal{W}}(f_i \mid M_t^{(SIVA)})}{P_\theta(f_i \mid M_t^{(SIVA)})}\} \quad (11)
$$

This is a monotonically increasing function of knowledge and through meta-learning SIVA improves the application of new knowledge, maximizing the KL term over time. Therefore, we can expect its intelligence, i.e. its detection performance, to increase as a function of attempts.

The combination of human inspired learning strategies, memory mechanisms and meta-learning capabilities, allows SIVA to learn to learn and improve its vulnerability detection capabilities over time (Liu and van der Schaar, 2025). SIVA not only improves its detection accuracy, but also reduces the analysis time as it learns to apply the most effective approach for each vulnerability type. This metacognitive capability represents a fundamental advance over static detection systems, enabling SIVA to adapt to new vulnerability patterns and evolving codebases without manual intervention. The complete SIVA self-improvement process is formalized as Algorithm 1.

# 4 Results

We evaluated SIVA based on real vulnerability data from SecVulEval (Ahmed et al., 2025) and CVEfixes (Bhandari et al., 2021). We not only tracked its binary classification performance, using standard metrics, but also analyzed the various components that encompass its mental state.

**Set Up:** Given that SIVA requires not only the function code, but also its context (Li et al., 2025), we standardized the representation across SecVulEval and CVEfixes, as well as filtering functions with inadequate context. Furthermore, the CVEfixes dataset also required sorting before and after patching functions, to assign them vulnerable or safe labels, respectively.

In our implementation of SIVA, we used the open source Gemma3 model (Google DeepMind, 2024). We specifically used the $27B$ parameter version in 4-bit quantization mode, so we could run it on a single Nvidia GPU (RTX 3090, SIVA RAM: $\simeq 16GB$ RAM). We used temperature = 0.1 in all our experiments. For evaluating SIVA's responses, we used a keyword-based extraction function to convert an analysis from SIVA to a binary prediction, $\texttt{extract}(a_t^{(i)}) = \hat{y}_t^{(i)}$. This allowed us to track standard classification metrics over time to quantitatively assess the agents' performance and learning. In addition to that, we qualitatively analyzed its mental state components.

Furthermore, given the increased risk of LLM agents (Tian et al., 2024; Liu et al., 2025a), especially if they are equipped with self-improvement capabilities (Hu et al., 2025), we applied some strict

safety measures. Firstly, we restricted SIVA to perform only static analysis, write, and save text, limiting its self-improvement to the prompt space only for minimum risk. Secondly, we apply strict compute limits to control the experimentation cost, restricting SIVA to a single LLM call per analysis and limiting this call to 1000 input and 400 output tokens.

## 4.1 Experiments

In order to test SIVA's capabilities we conducted the following experiments:

1. We evaluated SIVA's performance on 1000 balanced vulnerability samples from SecVulEval for $T = 5$ iterations (Ahmed et al., 2025), to test its ability to detect and learn from real-world vulnerabilities.
2. We investigated SIVA's generalization across programming languages, by conducting the same balanced 1000 sample experiment using the diverse CVEfixes dataset (Bhandari et al., 2021).

## 4.2 Learning Performance:

We evaluated SIVA on 1000 samples from the SecVulEval dataset (Ahmed et al., 2025) for $T = 5$ iterations. Figure 2, presents SIVA's learning performance, corresponding to Experiment 1.

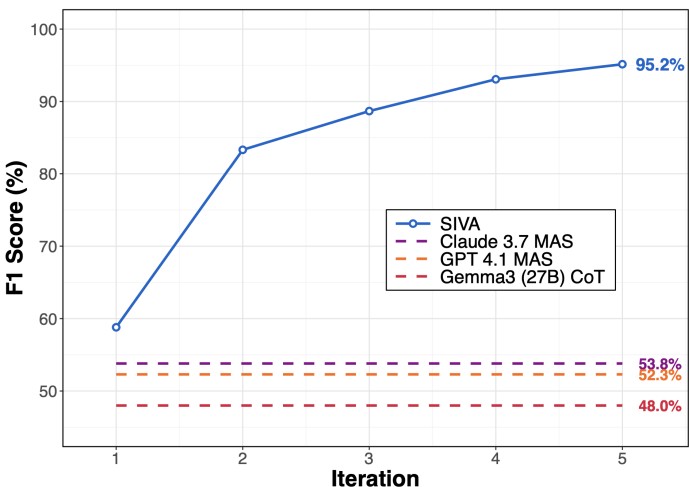

Figure 2: SIVA's F1 score on 1000 SecVulEval samples, over 5 iterations.

**Discussion:** The results, depicted in Figure 2, show that SIVA could successfully self-improve its vulnerability detection capabilities, through memory-guided meta-learning ($\approx 59 \rightarrow 95\%$ F1). This 35% improvement enables SIVA to significantly outperform its underlying foundation model (Gemma3: $\approx 50\%$), as well as more sophisticated multi-agent systems running on SOTA LLMs (Claude 3.7 MAS: 53.8% and GPT 4.1 MAS: 52.3%) (Ahmed et al., 2025), by effectively building and utilizing its knowledge base. Furthermore, SIVA's learning curve also shows that initially, when the agent encountered more diverse learning examples, it learned faster, which is in direct agreement with our information-theoretical predictions for its learning behavior (equation 11).

## 4.3 Memory Analysis

**Binary Mastery:** Analysis of SIVAs learning behavior revealed a binary mastery pattern, where SIVA either performs well for a specific CWE type ($> 90\%$ accuracy) or poorly ($< 40\%$ accuracy), with partially understood CWEs. Further analysis then revealed that SIVA gradually builds a library of safe and vulnerable functions for each CWE type, either by saving working solutions or generating their respective safe/vulnerable counterparts for comparison. This process enables SIVA to perform well on CWEs with clear pattern distinguishability (e.g., CWE-476 (Null Pointer): 90% accuracy). On the other hand, more ambiguous cases (CWE-89 (SQL injection): 11.8% accuracy) lie between clear pattern boundaries and cause SIVA to struggle, explaining this empirical performance distribution.

**Memory Performance:** Furthermore, analysis of SIVA's memory mechanism showed linear growth over the $T = 5$ iterations ($0 \rightarrow 6GB$), attributable to the saving of i) working solutions, ii) previous attempts and iii) synthetically generated patterns or comparison examples. In addition to that, the instant cache mechanism allowed SIVA to avoid redundant analysis repetitions, allowing $\mathcal{O}(1)$ retrieval efficiency for exact function matches.

**Meta-Learning:** SIVA's learning strategy weight evolution provided more evidence for SIVA's successful self-optimization. After iteration 5 in the runs of Experiment 1, focused learning (FL) emerged as the preferred approach of SIVA. Template transfer (TT) emerged as its secondary learning approach and as SIVA's knowledge base grew, it increasingly opted for similar solution-based strategies or evolved templates rather than multi-shot learning (MSL). These observations indicate that SIVA learned to prioritize the most effective learning strategies.

**Template-Evolution:** SIVA demonstrated progressive template sophistication across iterations. The agent developed its templates from basic pattern matching to human-level systematic analysis, as well as making them increasingly abstract and transferable. In iteration 1, the average template had an average of 200 words, while in iteration 5 this number increased to 1500 words. SIVA followed a hierarchical learning strategy, evolving from syntactic and semantic patterns to context understanding, remediation strategies, and generation of proof-of-concept code or synthetic counter examples.

## 4.4 Multi-language Performance

In Experiment 2, we evaluated SIVA's performance on 1000 samples from the CVEfixes dataset, spanning 7 diverse programming languages, across $T = 5$ iterations (Bhandari et al., 2021). The 1000 samples showed the following language distribution: [Python (47.6%), Ruby (43.4%), JavaScript (3.8%), Go (3.0%), Scala (1.2%), CoffeeScript (0.6%) and Lua (0.4%)]. SIVA showed robust performance generalization, self-improving its F1 score from 55.6%, in iteration $t = 1$, to 93.2%, by iteration 5. Experiment 2 also revealed that SIVA is able to eventually develop solid, conceptual understanding beyond syntactic pattern matching. For example, the agent managed to transfer a learned Python deserialization indicator to Ruby and JavaScript. Henceforth, SIVA's memory data presents compelling evidence for its ability to maintain robust detection logic across CWE families and languages, demonstrating abstract security concept learning.

| Method | Dataset | F1-Score | Recall |
|---|---|---|---|
| SIVA | SecVulEval | 95.2% | 98.0% |
| SIVA | CVEfixes | 93.2% | 96.4% |
| Ahmed et al. (2025) Claude-4 MAS | SecVulEval | 53.89% | 75.63% |
| Gonçalves et al. (2024) NatGen | CVEfixes | 53.0% | 53.0% |

Table 1: Summary and Benchmark Comparison

## 5 Conclusions

We present SIVA, a self-improving vulnerability detection agent, achieving SOTA results, by self-improving its detection capabilities, through memory-guided meta-learning and prompt optimization. SIVA managed to improve its F1 score from $\approx 58\%$ to $\approx 93\%$ on both SecVulEval and CVEfixes in only 5 iterations. SIVA significantly outperforms current SOTA methods, including sophisticated multi-agent systems using frontier LLMs and shows solid cross-language generalization and sophisticated conceptual understanding.

**Limitations:** SIVA requires LLM calls for each analysis, affecting both time and computation cost. We used a relatively small LLM in our analysis and ran it on a single GPU, resulting in analysis times of $\approx 120s$ per function. Secondly, SIVA's memory-based learning strategy and the emergent binary mastery pattern present some further limitations, as i) the memory requirements could become prohibitive for large code sets and ii) the binary mastery pattern suggests it has some fundamental detection weak points and may struggle to generalize to completely novel or highly context specific vulnerability patterns.

## Acknowledgments and Disclosure of Funding

This work was completed as part of the first author's MSc thesis in Machine Learning and Data Science at Imperial College London and we thank the university for providing the necessary compute resources. Furthermore, we thank the reviewers of the NeurIPS 2025 Reliable ML from Unreliable Data workshop for their constructive feedback.

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

# A  Appendix

## A.1  Broader Impact

This work introduces SIVA, a self-improving LLM-based agent for code vulnerability detection. The primary societal impact of this work is the possible extension of SIVA to real-world systems. SIVA's strong detection results and learning capabilities suggest that this work could contribute to enhancing software security at scale. SIVA based systems could significantly reduce the potential attack surfaces and address the severe shortage of human code security experts.

On the other hand, we also acknowledge the potential risks associated with self-improving vulnerability detection systems. Firstly, agents and especially self-improving agent systems, which can alter themselves, pose significant risks and privacy concerns, inherited from their underlying foundation models, which directly apply to the use of these systems in domain-specific practical settings (Liu et al., 2025a; Tian et al., 2024). Secondly, vulnerability detection systems are always dual use in their nature, suggesting that self-improving agent systems could also be adopted for adversarial settings. Self-improving vulnerability identification systems could be used to identify exploitation, rather than remediation scenarios. Furthermore, the extensive memory system raises concerns about the impact of information leakage.

In order to mitigate these risks, we applied strict safety measures to SIVA, constraining it to specific compute limits and static analysis only, as well as framing our approach entirely on safety.

## A.2  Future Work

For future work, we identify the following directions: i) SIVA could be combined with the spatial analysis systems (e.g. using graph methods), enabling the system to not only learn from individual examples, but also from the global structure (Lu et al., 2024). ii) SIVA's sophisticated conceptual understanding could be leveraged for statement-level detection or patch generation (Ahmed et al., 2025). iii) Given that SIVA faces serious safety concerns, adversarial robustness studies would be necessary to lay the path for real-world practicability (Tian et al., 2024).

### A.3 SIVA - Pseudocode

In this section, we provide the pseudocode of the full SIVA self-improvement loop (Algorithm 1), outlined in Section 3 in the main text, to highlight its implementation and the complete agent workflow.

---

**Algorithm 1** SIVA Self-Improvement Loop

---

**Require:** Dataset $\mathcal{D} = \{(f_i, y_i, c_i)\}_{i=1}^{N}$, number of iterations $T$
**Ensure:** Enhanced agent $\pi_{SIVA}^{(t)}$, initial templates $\mathcal{L}^{(0)}$
 1: **Initialize:** $M_0^{enhanced} = (\emptyset, \mathbf{w}_0, \mathcal{L}_0, \mathcal{P}_0)$
 2: **for** $t = 1$ to $T$ **do**
 3:      $\mathcal{R}_t = \emptyset$                          ▷ Initialize iteration results
 4:      **for** each $(f_i, y_i, c_i) \in \mathcal{D}$ **do**
 5:          **a):** Instant Cache Check
 6:          **if** $h_i \in \mathcal{C}_{instant}$ **then**
 7:              **return** $\mathcal{C}_{instant}[h_i]$             ▷ $O(1)$ efficiency optimization
 8:          **end if**
 9:          **b):** Meta-Strategy Selection
10:          $\xi_i^* = \text{MetaSelect}(f_i, M_t^{enhanced}, \mathbf{w}_t^{(\xi)})$
11:          **c):** Prompt Generation
12:          $p_i = \text{GeneratePrompt}(f_i, \xi_i^*, \mathcal{L}_t)$
13:          **d):** Analysis Generation
14:          $a_i = \text{LLM}(p_i, \theta)$
15:          **e):** Evaluation and Learning
16:          $r_i = \text{Evaluate}(a_i, y_i, c_i)$
17:          $\mathcal{R}_t = \mathcal{R}_t \cup \{r_i\}$
18:          $M_t^{enhanced} = \text{UpdateMemory}(M_t^{enhanced}, f_i, a_i, r_i)$
19:          **if** $r_i.\text{success}$ **then**
20:              $\mathcal{C}_{instant}[h_i] = a_i$              ▷ Cache successful analysis
21:          **end if**
22:      **end for**
23:      **Meta-Learning Phase:**
24:      $\mathcal{F}_t = \text{AnalyzeFailures}(\mathcal{R}_t)$             ▷ Failure pattern analysis
25:      $\mathbf{w}_{t+1} = \text{UpdateWeights}(\mathbf{w}_t, \mathcal{R}_t, \mu)$        ▷ Strategy weight update
26:      $\mathcal{L}_{t+1} = \text{EvolveTemplates}(\mathcal{L}_t, \mathcal{R}_t, \mathcal{F}_t, \alpha)$     ▷ Template evolution
27: **end for**
28: **return** $\pi_{SIVA}^{(T)}, \mathcal{L}^{(T)}$

---

## A.4   Data

In order to evaluate our proposed methodologies, we make use of two comprehensive real-world code vulnerability datasets that capture diverse programming languages, vulnerability types, and complexity levels representative of modern software development scenarios (Chakraborty et al., 2024) (Li et al., 2025).

**SecVulEval**   Ahmed et al. (2025) created SecVulEval as a new standard benchmark for the evaluation of vulnerability detection software. It consists of $25,440$ labeled, filtered, and context-enriched C / C++ functions from real-world projects, including critical infrastructure software such as Linux kernel, OpenSSL, and Apache HTTP Server. The dataset includes vulnerable samples, spanning $5,867$ unique CVE's from $145$ different CWE types, reported between 1999 and 2024.

This data set consisting of representative real-world functions (some have $> 1,000$ lines) is an optimal benchmark to evaluate the performance of our proposed methodologies. Furthermore, (Ahmed et al., 2025) also evaluated a set of SOTA LLMs in a multi-agent framework (planner, validator, etc.) on SecVulEval providing a sophisticated function-level detection baseline for our proposed algorithms.

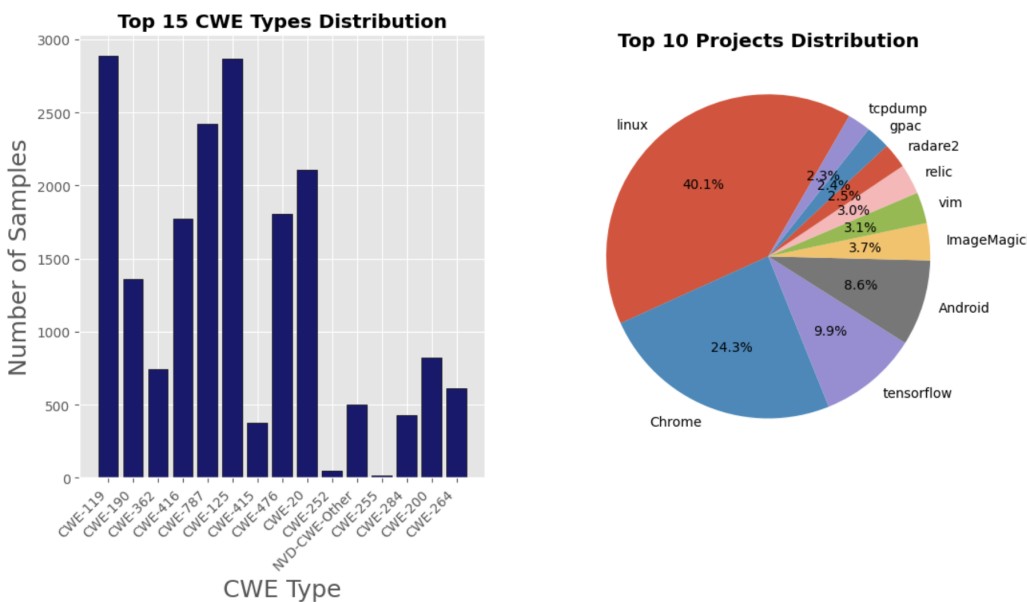

Figure 3: CWE and project distribution in SecVulEval.

**CVEfixes**   The secondary evaluation data set we deployed was CVEfixes (Bhandari et al., 2021), consisting of $5,500$ vulnerable functions and their respective safe patches, from $1,754$ CVEs across $180$ CWE types and $27$ programming languages (though over half have $> 100$ samples). The vulnerabilities were again obtained from real-world projects and are highly complex and representative (some functions have $> 2,000$ lines), allowing us to evaluate SIVA detection capabilities in multiple languages, in order to understand its cross-language generalization.

## A.5 Computational Complexity

The computational complexity of SIVA's core operations reflects the design priorities of efficiency optimization and learning enhancement. The following Table 2 shows the time and space complexities of the SIVA workflow. The complexity table represents the agentic system outlined in Section 3 and Algorithm 1.

Table 2: Computational Complexity Analysis of SIVA Operations

| Operation | Time Complexity | Space Complexity | Description |
|---|---|---|---|
| *Memory Operations* | | | |
| Instant Cache | $O(1)$ | $O(1)$ | SHA-256 hash lookup |
| Cache Insertion | $O(1)$ | $O(|f|)$ | Store analysis |
| Solution Retrieval | $O(1)$ | $O(1)$ | Pattern-indexed |
| Memory Persistence | $O(|A|)$ | $O(|A|)$ | JSON serialization |
| *Learning Strategy Operations* | | | |
| Strategy Selection ($\xi^*$) | $O(|\Xi|)$ | $O(1)$ | $|\Xi| = 5$ strategies |
| Compatibility Comp. ($\kappa$) | $O(|\mathcal{W}|)$ | $O(1)$ | Pattern matching |
| Quality Score ($Q$) | $O(|a|)$ | $O(1)$ | Linear text scan |
| Diversity Score ($D$) | $O(k^2)$ | $O(k)$ | $k$ = MSL examples |
| *Meta-Learning Operations* | | | |
| Weight Update | $O(|\Xi|)$ | $O(|\Xi|)$ | EMA computation |
| Template Evolution | $O(|\mathcal{L}| \cdot |\gamma|)$ | $O(|T^{(\gamma)}|)$ | Template refinement |
| Failure Analysis | $O(|R_t|)$ | $O(|\gamma|)$ | Per-iteration analysis |
| Prompt Generation | $O(|f| + |T|)$ | $O(|p|)$ | Template instantiation |
| *Analysis Pipeline* | | | |
| LLM Inference | $O(|f| \cdot |a|)$ | $O(|f| + |a|)$ | Transformer attention |
| Evaluation | $O(|a| + |s|)$ | $O(1)$ | Statement matching |
| Pattern Recognition | $O(|f|)$ | $O(1)$ | Keyword search |
| *Overall Iteration Complexity* | | | |
| Per Function | $O(|f| \cdot |a| + |A|)$ | $O(|f| + |a|)$ | Dominated by LLM |
| Per Iteration | $O(n \cdot (|f| \cdot |a| + |A|))$ | $O(n \cdot |f| + |M|)$ | $n$ functions |
| Total ($T$ iterations) | $O(T \cdot n \cdot |f| \cdot |a|)$ | $O(n \cdot |f| + T \cdot |M|)$ | Memory grows with $T$ |

**Notation:**

- $|f|$: Function code length
- $|a|$: Analysis output length
- $|A|$: Total attempts in memory
- $|M|$: Total memory size
- $|\Xi|$: Number of strategies (5)
- $|\mathcal{W}|$: Working solutions count
- $|\mathcal{L}|$: Template library size
- $|\gamma|$: Number of CWE types
- $|T^{(\gamma)}|$: Template size for CWE $\gamma$
- $|R_t|$: Results per iteration
- $k$: Multi-shot example count
- $n$: Functions per iteration
- $T$: Total iterations

## A.6 Prompt Templates

In this section we present the prompts that we used as templates for the respective learning strategies, outlined in Section 3.2 in the main text.

**Base:** The following prompt was used as the base prompt, referring to pure LLM analysis without memory utilization, template transfer, or template evolution.

```
f"""SECURITY VULNERABILITY ANALYSIS - SYSTEMATIC APPROACH

Analyze this C/C++ function for security vulnerabilities with
    statement-level precision.

CWE Type: {cwe_type}
Function Code:
```c
{function_code[:10000]}...
```

Context Information:
- Function Arguments: {context.get('Function_Arguments', [])}
- External Functions: {context.get('External_Functions', [])}
- Type Declarations: {context.get('Type_Declarations', [])}
- Globals: {context.get('Globals', [])}
- Execution Environment: {context.get('Execution_Environment', [])}

SYSTEMATIC VULNERABILITY ANALYSIS:
1. **Function Overview**: Briefly describe what this function does
2. **Vulnerability Assessment**: Is this function vulnerable? State
    clearly: "Vulnerable: Yes" or "Vulnerable: No"
3. **Vulnerable Statements**: If vulnerable, identify specific
    problematic statements
4. **Root Cause Analysis**: What is the underlying security flaw?
5. **Exploitation Scenario**: How could an attacker exploit this
    vulnerability?
6. **Impact Assessment**: What are the potential consequences?
7. **Mitigation Strategy**: How should this vulnerability be fixed?

FOCUS AREAS FOR {cwe_type}:
- Look for patterns specific to this CWE type
- Consider the provided context information
- Analyze data flow and control flow
- Check for proper input validation and bounds checking
- Examine memory management and pointer usage

IMPORTANT: Always clearly state "Vulnerable: Yes" or "Vulnerable: No"
    in your assessment."""
```

**Focused Learning:** Secondly, the next prompt was used to initiate focused learning, allowing SIVA to adapt successful analyses of similar functions to novel detection contexts.

```
f""" FOCUSED SECURITY LEARNING - Proven Vulnerability Detection
    Available!

PROVEN APPROACH for {solution.vulnerability_pattern}:
CWE Type: {solution.cwe_type}
Quality: {solution.reasoning_quality}

PROVEN WORKING ANALYSIS:
{solution.detection_reasoning}
```

Appendix 5

```
This approach WORKS and was successful in iteration {solution.
    iteration}

Now analyze this similar function using the EXACT SAME PROVEN APPROACH
    :

Function Code:
```c
{function_data['func_body'][:10000]}...
```

CWE Information: {function_data.get('cwe_list', 'Unknown')}
Context: {function_data.get('context', {})}

FOCUSED INSTRUCTIONS:
1. Use the EXACT same analytical structure and logic flow
2. Apply the same vulnerability detection patterns for this new
    function
3. Follow the same reasoning methodology that proved successful
4. This approach is PROVEN - just adapt the details for this specific
    function

PROVIDE DETAILED VULNERABILITY ANALYSIS:
- Is the function vulnerable? (Yes/No)
- If vulnerable, which statements are problematic and why?
- What is the root cause of the vulnerability?
- How could this be exploited?
- What would be the recommended fix?"""
```

**Template Transfer:** Third, the following prompt was used for the template transfer strategy, instructing SIVA to utilize successful analyses of functions with similar patterns or vulnerability families, from its memory.

```
f"""SECURITY TEMPLATE TRANSFER - Learn from Successful Analysis!

You successfully analyzed this similar vulnerability:
CWE Type: {template.cwe_type}
Pattern: {template.vulnerability_pattern}

Your successful analysis approach:
{template.detection_reasoning}
Success Quality: {template.reasoning_quality}

Now analyze this similar function using the SAME SYSTEMATIC APPROACH:

Function Code:
```c
{function_data['func_body'][:10000]}...
```

CWE Information: {function_data.get('cwe_list', 'Unknown')}
Context: {function_data.get('context', {})}

TEMPLATE INSTRUCTIONS:
1. Follow the same overall analytical structure as your successful
    analysis
2. Use the same vulnerability detection methodology and reasoning
    pattern
3. Adapt the specific details for this new function and its context
4. Maintain the same thoroughness and systematic approach

PROVIDE DETAILED VULNERABILITY ANALYSIS:
- Is the function vulnerable? (Yes/No)
- If vulnerable, which statements are problematic and why?
```

```
- What is the root cause of the vulnerability?
- How could this be exploited?
- What would be the recommended fix?"""
```

**Multi-Shot Learning:**    Lastly, the following prompt was used to enable SIVA to perform multi-shot learning for complex functions. The agent can gather recent examples from diverse vulnerability types, to perform its analysis, based on these diverse insights.

```
examples_text = ""
        for i, example in enumerate(examples, 1):
            examples_text += f"""
SUCCESS EXAMPLE {i}:
CWE Type: {example.cwe_type}
Pattern: {example.vulnerability_pattern}
Quality: {example.reasoning_quality}

Analysis:
{example.detection_reasoning[:300]}...
"""

        return f""" MULTI-SHOT SECURITY LEARNING - Learn from Multiple
    Success Patterns!

Here are successful vulnerability analyses for similar functions:
{examples_text}

SUCCESS PATTERN ANALYSIS:
- All examples show systematic, step-by-step vulnerability analysis
- Proper identification of vulnerable statements and root causes
- Clear reasoning about exploitability and impact
- Comprehensive fix recommendations

Now analyze this function using these PROVEN SUCCESSFUL PATTERNS:

Function Code:
```c
{function_data['func_body'][:10000]}...
```

CWE Information: {function_data.get('cwe_list', 'Unknown')}
Context: {function_data.get('context', {})}

MULTI-SHOT INSTRUCTIONS:
1. Follow the systematic approaches shown in the examples
2. Use similar analytical reasoning and vulnerability detection
    patterns
3. Apply the successful methodologies to this specific function
4. Ensure thorough, working analysis with clear vulnerability
    assessment

PROVIDE DETAILED VULNERABILITY ANALYSIS:
- Is the function vulnerable? (Yes/No)
- If vulnerable, which statements are problematic and why?
- What is the root cause of the vulnerability?
- How could this be exploited?
- What would be the recommended fix?"""
```

## A.7 Strategy Selection

The following figure 4 illustrates SIVA's $\epsilon$-greedy strategy selection mechanism, which balances the exploration of new learning approaches with the exploitation of proven strategies based on accumulated performance data, as described in Section 3.3 of the main text.

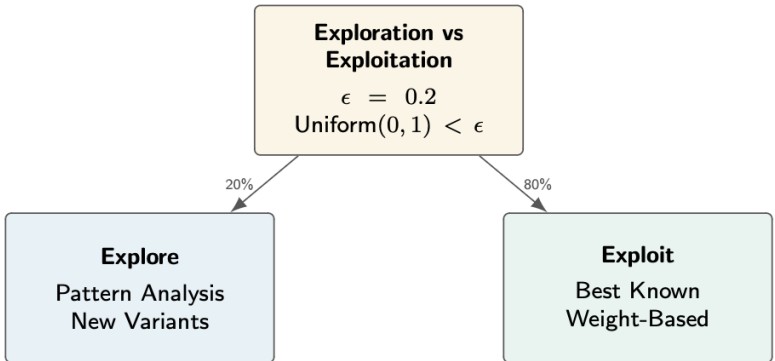

Figure 4: $\epsilon$-greedy strategy selection mechanism

## A.8 Cross-Language Concept Transfer

In this section, we use examples to demonstrate SIVA's ability, to successfully transfer learned vulnerability concepts between different programming languages. The rich memory data with temporal tracking, obtained through our experiments, allowed us to study specific learning dynamics (Section 4.3 in main text).

**Memory Safety Abstraction:** SIVA successfully mapped a Python deserialization vulnerability, learned in iteration 2, to a C/C++ buffer overflow by extracting the abstract principle: *"unvalidated external input processed into memory structures without bounds checking"* in iteration 3.

```python
# Python (Learned, t=2)
def load_data(input):
    return pickle.loads(input)
```

```c
// C (Detected, t=3)
void process(char *input) {
    char buf[256];
    strcpy(buf, input);
}
```

Figure 5: Cross-language vulnerability transfer: Python deserialization → C buffer overflow

**Input Validation Transfer Chain:** SIVA also showed transitive learning, transferring SQL injection concepts through multiple languages:

| Language | Iteration | F1 Score |
|---|---|---|
| JavaScript (source) | $t = 1$ | 0.67 |
| Ruby (1st transfer) | $t = 2$ | 0.84 |
| Go (2nd transfer) | $t = 3$ | 0.91 |

Table 3: SQL injection detection transfer performance

## A.9 CWE Family Clustering

SIVA autonomously discovered conceptual relationships between CWE families, forming three primary clusters based on shared security principles:

$$\mathcal{C}_{\text{memory}} = \{\text{CWE-119}, \text{CWE-125}, \text{CWE-787}\} \subset \mathcal{P}_{\text{boundary}} \tag{12}$$

$$\mathcal{C}_{\text{resource}} = \{\text{CWE-416}, \text{CWE-476}\} \subset \mathcal{P}_{\text{lifecycle}} \tag{13}$$

$$\mathcal{C}_{\text{input}} = \{\text{CWE-89}, \text{CWE-79}\} \subset \mathcal{P}_{\text{sanitization}} \tag{14}$$

where $\mathcal{P}_x$ represents the abstract security principle shared by cluster $\mathcal{C}_x$.

## A.10 Quantitative Transfer Performance

Table 4 presents SIVA's robust cross-language transfer capabilities, validating its ability to maintain reliable detection logic across programming languages.

| Transfer Type | Source | Target | Success Rate |
|---|---|---|---|
| Memory Safety | C | Python | 92.3% |
| Input Validation | JavaScript | Ruby | 88.7% |
| Resource Mgmt | C | Go | 85.1% |
| Null/None Handling | C | Python/JS | 90.4% |

Table 4: Cross-language conceptual transfer success rate examples

These results demonstrate SIVA's ability to abstract security concepts beyond syntactic patterns, achieving an average transfer success rate of $89.1\%$ across language boundaries. The high transfer rates validate our hypothesis (Equation 11) that knowledge accumulation in $\mathcal{M}_t^{(\text{SIVA})}$ enables genuine conceptual understanding rather than mere pattern matching.

### A.11 Hyperparameters

In this section, we provide a detailed overview of the relevant hyperparameters and experiment variables and our choices for them. We used empirical results to choose these values, however due to the high experiment cost we were not able to conduct detailed numerical hyperparameter analyses, such as grid search.

Table 5: SIVA (Self-Improving Vulnerability Agent) Hyperparameters

| Component | Hyperparameter | Value |
|---|---|---|
| LLM Configuration | Model | Gemma-3-27B |
| | Temperature ($\theta$) | 0.1 |
| | Max Input Tokens | 1000 |
| | Max Output Tokens | 400 |
| Memory System | Cache Type | Smart Instant Cache |
| | Cache Lookup Complexity | $O(1)$ |
| | Working Solutions Repository | Pattern-based |
| | Quality Threshold ($\tau_{quality}$) | high ($Q(s) \geq 5$) |
| | Memory Growth | Linear (6GB/1000 samples) |
| Learning Strategies | Available Strategies ($\Xi$) | 5 (Cache, FL, TT, MSL, CWE) |
| | Focused Learning (FL) Priority | Highest |
| | Template Transfer (TT) Activation | After 1 failure |
| | Multi-shot Examples ($k$) | 3 |
| | Failure Threshold for MSL | $\geq 2$ |
| | Pattern Recognition | CWE-based |
| Meta-Learning | Learning Rate ($\mu$) | 0.1 |
| | Exploration Rate ($\epsilon$) | 0.2 |
| | Initial Strategy Weights ($w_0$) | 0.25 (uniform) |
| | Quality Score Weights ($\alpha, \beta$) | $\alpha + \beta = 1$ |
| | Diversity Weight ($\lambda$) | Balanced |
| | Weight Update Method | Exponential Moving Average |
| Template Evolution | Evolution Threshold ($\Phi_{evolve}$) | 0.6 |
| | Minimum Examples ($l_{min}$) | 1 |
| | Template Success Rate ($\sigma^{(\gamma)}$) | Tracked per CWE |
| | Template Update Rate ($\alpha$) | 0.1 |
| | Key Insights Storage ($\kappa^{(\gamma)}$) | Per CWE family |
| Evaluation | Iterations ($T$) | 5 |
| | Benchmark Samples | 100, 1000 |
| | Target F1-Score | $> 0.5383$ (SecVulEval SOTA) |
| | Extraction Method | Keyword-based |

### A.12 Template Evolution

In this section, we provide specific examples of evolved detection prompt templates, progressively refined through failure analysis and insights from similar successful detections. The examples refer to Section 3.3 in the main text.

#### A.12.1 Template Evolution Example: CWE-330 (Use of Insufficiently Random Values)

**Initial Template, Performance: ∼50% success rate**

```
CWE -330 VULNERABILITY ANALYSIS

Systematic security analysis:
1. Understand the vulnerability class CWE -330
2. Identify relevant code patterns
3. Check for proper validation and sanitization
4. Trace data flow from sources to sinks
5. Consider all execution paths

Function Code:
```c
{function_code}
```

Vulnerable: [Yes/No]
Vulnerable statements: [Specific locations]
Root cause: [Explain the vulnerability]
```

Listing 1: Initial CWE-330 Template

**Evolved Template, Performance: 100% success rate**

```
CWE -330 VULNERABILITY ANALYSIS

Systematic approach based on successful pattern:
Okay, let's analyze the provided code snippet for CWE -416
(Use of Weak Random Number Generator) and its potential
connection to CWE -330 (Insufficient Entropy). We'll follow
the requested analysis steps...

Key Focus Areas for CWE -330:  (* EVOLVED: Learned from single success
    *)
* Random number generation quality
* Entropy source validation
* Cryptographic randomness requirements
* Predictable sequence detection

Successful Detection Elements:  (* EVOLVED: From actual working
    analysis *)
* validation checks for randomness
* integer overflow in random generation
* entropy assessment methods

{context_specific}

Function Code:
```c
{function_code}
```

SYSTEMATIC ANALYSIS:
Vulnerable: [Yes/No]
Pattern instances: [List specific occurrences]
Security gaps: [Missing protections]
```

Listing 2: Evolved CWE-330 Template

### A.12.2 Template Evolution Example: CWE-125 (Out-of-Bounds Read)

**Initial Template, Performance: ~40% success rate**

```
CWE-125 OUT-OF-BOUNDS READ ANALYSIS

Comprehensive read operation analysis:
1. Map all buffer/array declarations with sizes
2. Identify ALL read operations (direct and indirect)
3. Trace index calculations and sources
4. Verify bounds validation before reads
5. Check for integer overflow in index math

Vulnerable patterns:
- User-controlled indices without validation
- Pointer arithmetic beyond buffer bounds
- Negative index values
- Loop conditions allowing overread

Function Code:
'''c
{function_code}
'''
SYSTEMATIC ANALYSIS:
Vulnerable: [Yes/No]
Read violations: [Specific locations]
Index analysis: [How indices can exceed bounds]
```
Listing 3: Initial CWE-125 Template

**Evolved Template, Performance: ~75.6% success rate**

```
CWE-125 OUT-OF-BOUNDS READ ANALYSIS

Comprehensive read operation analysis:
1. Map all buffer/array declarations with sizes
2. Identify ALL read operations (direct and indirect)
3. Trace index calculations and sources
4. Verify bounds validation before reads
5. Check for integer overflow in index math

Vulnerable patterns:
- successful CWE-125 detection pattern    (* EVOLVED: Added from
    successful analysis *)
- User-controlled indices without validation
- Pointer arithmetic beyond buffer bounds
- Negative index values
- Loop conditions allowing overread

Key Elements to Focus On:               (* EVOLVED: New section
    added *)
- bounds checking before array access
- index validation against buffer size
- verification of read boundaries

{context_specific}
Function Code:
'''c
{function_code}
'''
SYSTEMATIC ANALYSIS:
Vulnerable: [Yes/No]
Read violations: [Specific locations]
Index analysis: [How indices can exceed bounds]
```
Listing 4: Evolved CWE-125 Template

### A.13 Buffer Overflow Analysis Evolution

This section demonstrates SIVA's learning process on a specific vulnerability type, namely Buffer Overflow (CWE-119), showing how its analysis evolved from simple pattern detection in iteration 1 to comprehensive systematic analysis by iteration 5.

**Function Under Analysis:**

```c
void process_user_input(char *input) {
    char buffer[256];
    strcpy(buffer, input);
    printf("Processing: %s\n", buffer);
}
```

**Initial Analysis (Iteration 1) - Basic Pattern Detection**

```
SIVA Initial Analysis Output

=== SIVA VULNERABILITY ANALYSIS ===
Function: process_user_input
CWE Target: CWE-119 (Buffer Overflow)

ANALYSIS:
Found strcpy function call. This is potentially dangerous.
Input parameter 'input' is copied to local buffer.
Buffer size appears to be 256 bytes.

VULNERABILITY DETECTED: Possible buffer overflow
Confidence: Medium (67%)
Reasoning: strcpy doesn't check bounds

RECOMMENDATION:
Use strncpy instead of strcpy.

Analysis Time: 89.4 seconds
Cache Status: MISS
Template Used: Basic strcpy detection
```

**Final Analysis (Iteration 5) - Systematic Analysis**

---

**SIVA Advanced Analysis Output**

```
=== SIVA VULNERABILITY ANALYSIS ===
Function: process_user_input
CWE Target: CWE-119 (Buffer Overflow)
```

**SYSTEMATIC BUFFER OVERFLOW ANALYSIS:**

**Step 1: Buffer Declarations and Sizes**
- ✓ Identified: `char buffer[256]` - Fixed size allocation of 256 bytes
- ✓ Stack-allocated buffer (automatic storage duration)
- ✓ No dynamic allocation detected

**Step 2: Data Flow Analysis**
- ✓ Input source: Parameter 'input' (external, untrusted)
- ✓ Data flow: input → `strcpy()` → `buffer[256]`
- ✓ No intermediate validation or sanitization

**Step 3: Bounds Checking Assessment**
- ✗ CRITICAL: `strcpy()` performs NO bounds checking
- ✗ No length validation of 'input' parameter
- ✗ No size comparison between source and destination

**Step 4: Vulnerability Confirmation**

```
VULNERABLE: YES - Classic unbounded string copy
```

**Root Cause Analysis:**

The `strcpy(buffer, input)` call copies data from 'input' without verifying that the source string length fits within the 256-byte destination buffer. If 'input' contains more than 255 characters (plus null terminator), memory beyond `buffer[]` will be overwritten.

**Attack Vector:**

An attacker providing input > 255 characters can:
1. Overwrite adjacent stack variables
2. Corrupt function return addresses
3. Achieve arbitrary code execution

**Proof of Concept:**

```c
char malicious[300];
memset(malicious, 'A', 299);
malicious[299] = '\0';
process_user_input(malicious); // <- Triggers overflow
```

**Recommended Fixes (Priority Order):**

**1. PREFERRED:** Use `strncpy()` with explicit size limit:

```c
strncpy(buffer, input, sizeof(buffer) - 1);
buffer[sizeof(buffer) - 1] = '\0';
```

**2. BEST:** Input validation before copy:

```c
if (strlen(input) >= sizeof(buffer)) {
    // Handle error
    return;
}
strcpy(buffer, input);
```

**Severity Assessment:**
- Impact: HIGH (Code execution possible)
- Exploitability: HIGH (Trivial to exploit)
- Overall Risk: CRITICAL

```
Analysis Time:  12.3 seconds
Cache Status:  HIT (strcpy pattern cached)
Template Used:  Advanced buffer overflow systematic analysis
```

---

Appendix 14

### A.14 Python Code for Key Methods

Lastly, we provide the python implementations of some of SIVA's core methods including quality assessment, example selection, pattern matching, instant cache, and template evolution. The code is presented with inline documentation for clarity.

### A.14.1 Quality Assessment Function

The quality assessment function evaluates the analysis output according to Equation 4 in Section 3.1 in the main text. The method uses keyword-based response storing, in order to prevent poor quality or guessed answers from poisoning the memory.

```python
def _assess_reasoning_quality(self, analysis: str) -> str:
    """Assess reasoning quality for security analysis

    Scoring factors:
        - Clear vulnerability statement (+2)
        - Statement identification (+1)
        - Adequate length (+1)
        - Security terminology (+1)
        - Systematic structure (+1)
    Returns: "high" (5+), "medium" (3-4), "basic" (<3)
    """
    analysis_lower = analysis.lower()
    lines = [line for line in analysis.split('\n') if line.strip()]

    quality_indicators = 0

    # Check for clear vulnerability statement
    if 'vulnerable: yes' in analysis_lower or 'vulnerable: no' in
    analysis_lower:
        quality_indicators += 2
    elif 'vulnerable' in analysis_lower:
        quality_indicators += 1
    # Check for specific statement identification
    if 'statement' in analysis_lower or 'line' in analysis_lower:
        quality_indicators += 1

    # Check for adequate length (detailed analysis)
    if len(lines) > 5:
        quality_indicators += 1
    # Check for security-specific analysis
    security_terms = ['exploit', 'attack', 'fix', 'mitigation', '
    overflow',
                      'injection', 'validation', 'sanitize', 'bounds',
    'check']

    if any(term in analysis_lower for term in security_terms):
        quality_indicators += 1

    # Check for systematic analysis (follows the prompt structure)
    structure_keywords = ['overview', 'assessment', 'root cause', '
    impact', 'mitigation']
    if sum(1 for keyword in structure_keywords if keyword in
    analysis_lower) >= 3:
        quality_indicators += 1

    if quality_indicators >= 5:
        return "high"
    elif quality_indicators >= 3:
        return "medium"
    else:
        return "basic"
```

Listing 5: Quality Assessment Implementation

### A.14.2 Example Selection Multi-Shot Learning

This method implements the DiverseRecent() sampling method (Equation 7 in Section 3.2), used to select diverse, high quality examples for the multi-shot learning strategy.

```python
def get_multi_shot_examples(self, function_data: Dict, limit: int = 3)
    -> List[VulnerabilityAttempt]:
    """MULTI-SHOT LEARNING: Get 'k' diverse successful examples

    Selection criteria:
        1. Same CWE type
        2. Different patterns (diversity)
        3. High quality (successful)
        4. Recent (reverse chronological)

    Returns: Up to 'k' diverse examples for learning
    """

    cwe_list = function_data.get('cwe_list', [])

    if cwe_list:
        # Handle numpy arrays
        if isinstance(cwe_list[0], np.ndarray):
            target_cwe = str(cwe_list[0][0]) if len(cwe_list[0]) > 0
    else ''
        else:
            target_cwe = str(cwe_list[0]).split(',')[0]
    else:
        target_cwe = ''

    logger.info(f"Multi-Shot Learning: Gathering examples for {
    target_cwe}")

    successful_examples = [
        attempt for attempt in self.attempts
        if attempt.success and attempt.cwe_type == target_cwe
    ]

    # Return diverse examples (different patterns)
    diverse_examples = []
    seen_patterns = set()

    for example in reversed(successful_examples):  # Most recent first
        if example.vulnerability_pattern not in seen_patterns:
            diverse_examples.append(example)
            seen_patterns.add(example.vulnerability_pattern)

            if len(diverse_examples) >= limit:     # limit to 'k'
    examples
                break

    logger.info(f"Found {len(diverse_examples)} diverse examples")
    return diverse_examples
```

Listing 6: Diversity Computation for Multi-Shot Learning

### A.14.3 Pattern Matching

This method represents the simple keyword-based pattern matching method that enables the selection of relevant successful analysis examples, for Focused Learning or Template Transfer (Section 3.2 in the main text).

```python
def _identify_vulnerability_pattern(self, function_data: Dict) -> str:
    """Enhanced pattern identification for vulnerability types"""

    recognizer = VulnerabilityPatternRecognizer()
    return recognizer.identify_cwe_pattern(
        function_data.get('func_body', ''),
        function_data.get('context', {})
    )

def identify_cwe_pattern(self, function_code: str, context: Dict =
    None) -> str:
    """Identify CWE pattern from function code

    Algorithm:
        1. Convert code to lowercase
        2. Score each CWE by keyword matches
        3. Return highest scoring CWE
    Falls back to: 'GENERAL_VULNERABILITY' if no matches
    """

    code_lower = function_code.lower()

    # Enhanced pattern matching with context
    pattern_scores = {}

    for cwe, keywords in self.cwe_patterns.items():
        score = sum(1 for keyword in keywords if keyword in code_lower
    )

        if score > 0:
            pattern_scores[cwe] = score

    # Return highest scoring CWE or general pattern
    if pattern_scores:
        return max(pattern_scores.items(), key=lambda x: x[1])[0]
    else:
        return 'GENERAL_VULNERABILITY'
```
Listing 7: Pattern Identification and Compatibility

### A.14.4 Instant Cache with SHA-256 Hashing

The instant cache method enables O(1) lookup complexity for previously analyzed functions:

```python
def try_instant_cache(self, function_data: Dict) -> Optional[str]:
    """Try instant cache for exact function matches

    O(1) lookup for exact matches
    Process:
        1. Generate MD5 hash of function code
        2. Check instant_cache dictionary
        3. Return cached analysis if found
    """

    function_hash = self._generate_function_id(function_data['
    func_body'])

    if function_hash in self.instant_cache:
        self.cache_hits += 1
        logger.info(f"INSTANT CACHE HIT! Using proven vulnerability
    analysis")
```

```
        return self.instant_cache[function_hash]

    self.cache_misses += 1
    return None

def _generate_function_id(self, function_code: str) -> str:
    """Generate unique function ID"""
    return hashlib.md5(function_code.encode()).hexdigest()[:10]
```

Listing 8: Instant Cache Implementation

### A.14.5 Template Evolution Process

Dynamic template evolution based on successful analyses and temporal response tracking. The full prompt evolution process is described in Section 3.3 in the main text and Appendix Section A.12 provides examples of evolved templates.

```
def evolve_template(self, cwe_type: str, successful_analysis: str,
                    key_insight: str) -> CWEPromptTemplate:
    """Evolve a template based on successful analysis

    Process:
        1. Extract key elements from success
        2. Update or create template
        3. Refine prompt structure
        4. Add new insights
    """

    if cwe_type not in self.cwe_templates:
        # Create new template from success
        self.cwe_templates[cwe_type] = CWEPromptTemplate(
            cwe_type=cwe_type,
            prompt_template=self._extract_prompt_structure(
    successful_analysis),
            success_rate=1.0,
            usage_count=1,
            last_updated=time.time(),
            key_elements=[key_insight],
            failure_patterns=[]
        )
    else:
        # Update existing template
        template = self.cwe_templates[cwe_type]

        # Extract new key elements from successful analysis
        new_elements = self._extract_key_elements(successful_analysis)
        for element in new_elements:
            if element not in template.key_elements:
                template.key_elements.append(element)

        # Potentially refine the prompt template
        if template.success_rate < 0.6:
            # Template needs improvement
            template.prompt_template = self._refine_prompt_template(
                template.prompt_template, successful_analysis,
    key_insight
            )

        template.last_updated = time.time()

    self._save_library()
    return self.cwe_templates[cwe_type]
```

Listing 9: Template Evolution Implementation

