# OpenReview forum: "SIVA: Self-Improving Vulnerability Agent"
_NeurIPS.cc/2025/Workshop/Reliable_ML — NeurIPS 2025 - Reliable ML Workshop_

### Official Review · Reviewer_wWhg · 2025-09-17
**Impressive results.**

**Rating:** 8
**Confidence:** 2

**Review:**

**Summary.** This paper provides an interesting result on a self-improving agent for the task of detecting code vulnerability. The proposed agent, SIVA, uses a base large language model (Gemma-3 27B) and enhances its performance through a novel, iterative learning framework. The core of its self-improvement mechanism is *memory-guided meta-learning*, which dynamically optimizes prompts and selects learning strategies based on past performance. With several iterations, the agent can build a knowledge base of successful and failed attempts, using retrieval mechanisms to inform its analysis of new code snippets. Experiments show taht SIVA improves its F1 score on a real-world vulnerability dataset from an initial 58% to 95% in just five iterations, substantially outperforming state-of-the-art multi-agent systems. Furthermore, it demonstrates strong cross-language generalization capabilities.

**Strengths.** Novel ideas and significant improvements, which outperform even the most advanced models like GPT 4.1.

**Weaknesses.** The main experiment is only run once, so there is a lack of measure of variance. Moreover, this framework seems to be substantial computational and space costs, potentially raising concerns on scalability.

**Suggestions.**
More background on the equations would be nice: define what $\mathcal W$, $M_{memory} and $x$ are before presenting equation (1). The notion of $U = \mathcal W / M_{memory}$ is also somewhat confusing without stating explicitly what it means.

---

### Official Review · Reviewer_zjs9 · 2025-09-20
**Review for SIVA: Self-Improving Vulnerability Agent**

**Rating:** 7
**Confidence:** 1

**Review:**

## Summary

This paper presents SIVA, the first (to the authors’ knowledge) self-improving LLM-based agent for vulnerability detection. Using a tiered memory system, meta-learning, and dynamic CWE-specific prompts, SIVA adapts over time and achieves large performance gains: F1 improves from 58% to 95% on SecVulEval and 93% on CVEfixes, outperforming both single-model and multi-agent baselines.

## Strengths:
- The authors applies LLM to an important yet specialized field of vulnerability detection. The paper’s approach has clear practical use cases.
- Relatively clear explanations of how their method works, grounded in information-theoretic learning.
- The author acknowledged the dual-use nature of vulnerability-detection agents: they can be used for adversarial settings to identify exploitation.
- Did not just report the metrics, but also shown real examples of prompts evolve in sophistication

## Weakness / Suggestions:
- Scalability of the method might be limited, given that it takes ≈120s per function
- Very minor, but in line 96, the “where” should be lowercase?
- Again, very minor, but I find the (automated) colors in the prompt segments in appendix to be a bit confusing (e.g. line 747 - line 749)